# On-Device Deep Learning Inference for Efficient Activity Data Collection

**DOI:** 10.3390/s19153434

**Published:** 2019-08-05

**Authors:** Nattaya Mairittha, Tittaya Mairittha, Sozo Inoue

**Affiliations:** Graduate School of Engineering, Kyushu Institute of Technology, 1-1 Sensui-cho, Tobata-ku, Kitakyushu-shi, Fukuoka 804-8550, Japan

**Keywords:** activity recognition, data collection, on-device deep learning inference, smartphone sensors, user feedback

## Abstract

Labeling activity data is a central part of the design and evaluation of human activity recognition systems. The performance of the systems greatly depends on the quantity and “quality” of annotations; therefore, it is inevitable to rely on users and to keep them motivated to provide activity labels. While mobile and embedded devices are increasingly using deep learning models to infer user context, we propose to exploit on-device deep learning inference using a long short-term memory (LSTM)-based method to alleviate the labeling effort and ground truth data collection in activity recognition systems using smartphone sensors. The novel idea behind this is that estimated activities are used as feedback for motivating users to collect accurate activity labels. To enable us to perform evaluations, we conduct the experiments with two conditional methods. We compare the proposed method showing estimated activities using on-device deep learning inference with the traditional method showing sentences without estimated activities through smartphone notifications. By evaluating with the dataset gathered, the results show our proposed method has improvements in both data quality (i.e., the performance of a classification model) and data quantity (i.e., the number of data collected) that reflect our method could improve activity data collection, which can enhance human activity recognition systems. We discuss the results, limitations, challenges, and implications for on-device deep learning inference that support activity data collection. Also, we publish the preliminary dataset collected to the research community for activity recognition.

## 1. Introduction

In the field of ubiquitous computing, researches on human activity recognition technology using mobile sensors such as smartphones have been conducted [1]. Smartphone-based activity recognition systems aimed at physical activities recognition such as walking or running, are based on mobile sensor data. The sensor data may be recorded directly on the subject such as by carrying smartphones that have accelerometers and gyroscopes [2]. Understanding what users are doing in the physical world allows the smartphone app to be smarter about how to interact with them. However, a central challenge in smartphone-based activity recognition is data annotation studies in order to assess the labels describing the current activity while this activity is still on-going or recent to ensure that the dataset is labeled correctly. The quality and quantity of annotations can have a significant impact on the performance of the activity recognition systems. Hence, it is unavoidable to rely on the users and to keep them motivated to provide labels. To overcome the challenge of self-labeling [3], we introduce the idea of utilizing on-device deep learning inference for optimizing activity data collection. The rapid performance increase of low-power processors and the huge demand of internet of things (IoT) applications brought new ways for deploying machine/deep learning models on edge devices. On-device machine learning by fusing the inertial sensors such as smartphones have been explored [4,5,6]. These findings allow the activity recognition system to be feasibly identify frequent behavioral patterns on edge devices; meanwhile, deep learning revolution in the field of machine learning tends to result in higher accuracy and performs exceptionally well on machine perception tasks on smaller devices with limited resources [7,8,9]. TensorFlow Lite [10] was designed to enable easy to perform machine/deep learning inference on mobile, embedded, and IoT devices with low latency and a small binary size, “at the edge” of the network, instead of sending data back and forth from a server. Thus, we will exploit the power of on-device deep learning to provide estimated activities on a smartphone in order to optimize activity data collection.

In this paper, we want to show that if we give estimated activities using on-device deep learning inference through notifications as feedback to users while they are requested for labeling, we can improve data annotation tasks for activity recognition systems. The novel idea works by the user getting estimated activities through notifications on a smartphone as feedback that motivates for efficient activity data collection. Estimated activities are automatically inferred by periodically reading short bursts of smartphone sensor data and processing them using on-device deep learning with a long short-term memory (LSTM) model [11] without the model retrained. To evaluate this contribution, we trained the model for on-device deep learning with the open dataset [12], conducted the experiments with proposed and traditional methods (see Table 1) to collect the evaluated dataset, validated the dataset collected using several machine/deep learning algorithms, and showed that our proposed method outperformed the traditional method. In summary, the contribution of the paper are listed as the following:We introduce a system design of integrating on-device deep learning inference and activity recognition. We describe on-device deep learning inference using an LSTM-based method which can be used for efficient activity data collection, where estimated activities are used as feedback through notifications on a smartphone.We present the proposed method where we provide estimated activities using on-device inference through notifications and the traditional method where we provide simple sentences without estimated activities through notifications. Our proposed method can be applied not only to LSTM but also other models for on-device inference. To evaluate in a realistic setting, we train the model used for on-device deep learning with the open dataset, implement a system and deploy the system to a laboratory, conduct the experiments, review and use the dataset obtained for evaluations.We evaluate the quality of the proposed method using a standard activity recognition chain by comparing the performance results of several machine learning algorithms as well as a deep learning algorithm with the traditional method. We show that when estimated activities using on-device inference are provided to users as feedback, we can improve the quality of data collection (e.g., the accuracy of several machine learning algorithms has improvements with the proposed method). We also compare the quantity of data collected between the proposed method and the traditional method by showing that the amount of data collected has increased with the proposed method.We discuss the results, limitations, challenges, and implications for on-device deep learning inference that support activity data collection and spark future studies.We also publish the preliminary dataset openly as Appendix A in this paper, which might be useful for activity recognition and the research community.

We will begin by setting this work in the context of existing research on activity recognition and on-device deep learning in Section 2. We will then describe our method, experimental setup, and experimental evaluation in depth in Section 3, Section 4 and Section 5. Finally, we will conclude with a discussion of the results and the future work derived from the knowledge obtained in this paper in Section 6 and Section 7.

## 2. Related Works

Data annotation is a challenging process and a major limitation to the development of activity recognition systems. The classification task and the training phase of models can be performed offline or online [13,14,15]. Collecting accurate labels (annotation) comes with a hefty price tag, in terms of human effort. Either to have the data labeled by third-party observers or self-labeling both are costly, time-consuming, tedious, and they have the risk of missing some of the activity labels. For instance, while employing observers to annotate labels, it is correct segments but costly [16]. By contrast, to get labels using self-labeling and experience sampling, it is lower cost, but incorrect segments [3]. Another method such as offline labeling it also takes a long time and privacy issues [17]. This is a general problem for all supervised learning methods, which not only require the presence of a big dataset but also require human supervision to annotate the dataset. There are a lot of works being carried out to support and facilitate the process of data annotation [18,19,20].

To make capital out of the cloud, we occasionally offload data on small devices such as smartphones and smartwatches to the cloud for storage and processing. For example, physical activity information derived from the accelerometers of wearables is often transferred to and stored in the cloud. The ability to offload complicated tasks from devices with limited computation capabilities to virtual process capacity in the cloud is interesting; however, there are some limitations, for instance, advances in hardware capabilities and privacy threats. Therefore, reducing the use of the cloud for a mobile application, several researchers have proposed mechanisms to substitute the cloud with local networks or compute without any network connection (so-called on-device data analysis) [21,22,23]. Since hardware and devices are becoming ever more capable while decreasing in size and weight through miniaturization [24], performing analysis on-device and maintaining data in personal storage are possible and appealing. Existing studies on ubiquitous and pervasive applications that could work without any network connection or by using a hybrid approach that does not send all data into the cloud has been increased. For instance, the authors of [4,5,6] implemented machine learning inference on edge devices and demonstrated that on-device processing could sometimes improve energy efficiency and response time relative to off-device processing. Also, the authors [7,8,9] showed the idea of implementing more complex and taxing algorithms such as deep learning on these small devices.

The application of deep learning for human activity recognition has been effective in extracting discriminative features from raw input sequences acquired from body-worn sensors. Researchers have been adopting deep learning methods for activity recognition [11,25,26]. Recognizing human activities user-independently on smartphones based on accelerometer data using LSTM networks are well-suited [27]. Thus, we will focus on an LSTM-based deep learning model for activity recognition in this study; however, there are lots of challenges on both steps in a scenario of complex data and lacking sufficient domain knowledge. While the authors of [27] proposed an LSTM-based feature extraction approach to recognize human activities using three-axial accelerometer data, and showed the method could achieve high accuracy. Previously, [28] pointed to recognizing multiple overlapping activities using an algorithm of a compositional CNN + LSTM. However, using an LSTM for on-device inference is deficient. The authors of [29] introduced RSTensorFlow: an accelerated deep learning framework on commodity android devices using the heterogeneous computing framework RenderScript. The authors of [30] explored optimizations to run recurrent neural network (RNN) models locally on mobile devices for activity recognition. While they point to the significant issues in the performance of running different models architectures for Android devices while running deep learning models, we will present differences both on applications and purpose. While many of their research questions were similar to ours in using an LSTM-based deep learning model for activity recognition in various application domains, yet, no one has studied the use of on-device deep learning inference as feedback for activity data collection by giving estimated activities are inferred through notifications, which we will present in this paper.

## 3. Methods

In this section, we provide a descriptive view of the proposed on-device deep learning inference for efficient activity data collection system. The architecture of this system is depicted in Figure 1. The system is composed of several technical building blocks including the following: (1) to build an LSTM-based deep learning model used for on-device inference; (2) to collect accelerometer sensor data and activity labels efficiently; and (3) to provide estimated activities as feedback through smartphone notifications for efficient data collection.

### 3.1. To Build an LSTM Model Used for On-Device Inference

In this section, we propose how to build an LSTM model used for on-device inference. We employ the open dataset provided by the wireless sensor data mining (WISDM) Lab [12] to build an activity recognition model for on-device deep learning inference. A schematic diagram of the proposed LSTM-based deep learning model for activity recognition system is depicted in Figure 2.

Firstly, let us describe the core idea behind LSTMs. The RNN dynamics can be described using deterministic transitions from previous to current hidden states. The deterministic state transition is a function
$$\begin{array}{c} \mathrm{RNN}:h_{t}^{l-1},h_{t-1}^{l}\to h_{t}^{l} \end{array}$$

For classical RNNs, this function is given by
$$\begin{array}{c} h_{t}^{l}=f\left(T_{n,n}h_{t}^{l-1}+T_{n,n}h_{t-1}^{l}\right),\;\mathrm{where}\;f\in \{\mathrm{sigm},\tanh\} \end{array}$$

The LSTM has complicated dynamics that allow it to simply “memorize” information for an extended number of timesteps. The “long term” memory is stored in a vector of *memory cells*
$c_{t}^{l}\in R^{n}$. Although many LSTM architectures that differ in their connectivity structure and activation functions, all LSTM architectures have explicit memory cells for storing information for long periods of time. The LSTM can decide to overwrite the memory cell, retrieve it, or keep it for the next time step. The LSTM architecture used in this peper is given by the following equations [31]:$$\begin{array}{c} \mathrm{LSTM}:h_{t}^{l-1},h_{t-1}^{l},c_{t-1}^{l}\to h_{t}^{l},c_{t}^{l}\\ \left(\begin{array}{c} i\\ f\\ o\\ g \end{array}\right)=\left(\begin{array}{c} \mathrm{sigm}\\ \mathrm{sigm}\\ \mathrm{sigm}\\ \tanh \end{array}\right)T_{2n,4n}\left(\begin{array}{c} h_{t}^{l-1}\\ h_{t-1}^{l} \end{array}\right)\\ c_{t}^{l}=f⊙c_{t-1}^{l}+i⊙\;g\\ h_{t}^{l}=o⊙\tanh\left(c_{t}^{l}\right) \end{array}$$

In these equations, sigm and tanh are applied elementwise.

We use the WISDM dataset mentioned to build an activity recognition model. The reason why we first build the model by employing an existing dataset, and we then utilize it for our proposed on-device inference method because we concern the issue that our system cannot draw any inferences for users since it has not yet gathered sufficient information. This problem usually occurs in computer-based information systems which involve a degree of automated data modeling. It is a well-known and well-researched problem, so-called the cold start problem [32]. The human activity recognition dataset built from the recordings of 29 subjects performing regular activities while carrying a waist-mounted smartphone with embedded inertial sensors. This dataset contains 1,098,207 examples and six attributes, including, user, activity, timestamp, x-acceleration, y-acceleration, z-acceleration without missing attribute, collected through controlled, laboratory conditions. There are 6 activity types of movement that we try to classify: walking (38.6%), jogging (31.2%), upstairs (11.2%), downstairs (9.1%), sitting (5.5%), standing (4.4%). The dataset’s description is detailed in [12].

An LSTM takes many input vectors to process them and output other vectors. In our case, the “many to one” architecture is used: we accept time series of feature vectors (one vector per time step) to convert them to a probability vector at the output for classification, as shown in Figure 3.

As we can see from Figure 2, the inputs are raw signals obtained from multimodal-sensors, which is a discrete sequence of equally spaced samples $\left(x_{1},x_{2},\ldots ,x_{T}\right)$, where each data point $x_{t}$ is a vector of individual samples observed by the sensors at time *t*. These samples are segmented into windows of a maximum time index *T* and fed into LSTM-based deep learning model. Each generated sequence contains 200 training with 3 input parameters (3-axis accelerometer) per time steps. The model is trained for a maximum of 50 epochs by two fully-connected and two LSTM layers (stacked on each other) with 64 units each. We use rectified linear units (ReLUs) for the hidden layers to increase the robustness of the model as well as remove any simple dependencies between the neurons preventing over fitting, and use the dropout technique to avoid overfitting in our model (Equation (Equation 1)), where a rectified linear unit has output 0 if the input is less than 0, and raw output otherwise.
(1)ReLU(x)=max(0,x).

Finally, the model outputs class prediction scores for each time step $\left(y_{1}^{L},y_{2}^{L},\ldots ,y_{T}^{L}\right)$, where $y_{t}^{L}\in R^{C}$ is a vector of scores representing the prediction for a given input sample $x_{t}$ and C is the number of activity classe, which are fed into the softmax layer to determine class membership probability.

Also, we use an optimization algorithm called Adam [33] to minimize the cost function by backpropagating its gradient and updating model parameter. The core hyper-parameters explored in this model are listed in Table 2.

For validating the trained model against test data, we apportion the data into training and test sets, with an 80–20 split. After each epoch of training, we evaluate the performance of the model on the validation set. We select the epoch that showed the best validation-set performance and apply the corresponding model to the test-set. As a result, we opt the final epoch that the accuracy and weighted F1-score both are reached over 97% (0.975 and 0.972, respectively) and loss is hovered at around 0.2. Note that the class distribution of the WISDM dataset has the sample imbalances among activity classes which can affect machine learning [34]. We could not collect more data that could balance our classes; however, we show the weighted F1-score for additional performance metric that is preferable if there is a class imbalance problem, not just only accuracy [35]. Since the smartphone is attached on the waist and each series to classify has just a 200 sample window, those predictions are extremely accurate. If we have a look at the confusion matrix of the model’s predictions in Figure 4, we can see that our model performs really well. Although we can see some notable exceptions that there are difficulties in making the difference between walking, upstairs and downstairs, the model is almost always able to identify the movement type on a smartphone correctly. The visual insight of the results are presented in Figure 4 and Figure 5.

### 3.2. To Collect Accelerometer Sensor Data and Activity Labels Efficiently

We requested participants to carry a waist-mounted Android smartphone (Wiko Tommy3 Plus (Android 8.1)) with embedded inertial sensors, install the mobile app on smartphones to select and record their daily life activities from the list of predefined labels. Information about the demography of participants and the duration of the experiment are reported in Section 4. The mobile app is extended from our work [36] called “**FahLog**”, as shown in Figure 6, when a user selects the activity in Figure 6a, the labels for each activity class will be put into the right column as shown in Figure 6d. Then the user has to record it by pushing the button to start and stop recording while they are carrying out the activity by following the steps as shown in Figure 6b–d. Each time the user taps an activity label box, it will transition to before start (▶) → doing activity (⊙) → finish (✓) so a user can record the start and end of the activity. Since another activity may be performed while performing one activity, multiple activity labels can be started and ended in parallel. The activity labels can then be uploaded to the server when it is connected to the network. Otherwise, data will be stored on the smartphone until there is internet access. Moreover, we capture sensors and activity labels through smartphones to recognize activities using smartphone sensors continuously. Hence, we set the sampling rate of the app for the ‘standard’ settings of Android programming API, which is the slowest setting, where they are sampled 200 milliseconds when they are not busy, then we take one minute time windows for calculating time windows, it is enough of a sampling rate for such data collection. Note that the FahLog annotation tool will be unexpired and can be applied for other activity data collection experiments such as crowdsourcing [37] or nursing cares [38]. Also, it provides on Google Play openly [39]. It can alter multiple activity types from our server configuration connected to the app that has already been set up [40] as well as it can modify the user interface of the tool for each specific purpose (e.g., in this paper, adding notifications for showing estimated activities as feedback). More features have been published in [36].

### 3.3. To Provide Estimated Activities as Feedback through Smartphone Notifications

We interpret the results that retrieve from model inference. We use a list of probabilities that the model returned. We then meaningfully map them to relevant categories (activity classes) and present it on mobile notification to the user. Figure 7 presents an example of the results that are displayed on a notification. Note that to prevent excessive interruptibility and to optimize resources, we stop activity reporting if the device has been still for a while, and use low power sensors to resume reporting when it detects changes in the user’s activity (e.g., changing from walking to running) with mean inference time of 2846.0 ms. Also, when we put the deep learning model on the device and use a battery monitor application for Android smartphones to monitor the battery level, it increases energy consumption on its smartphone by 5% on average compared with the traditional manner without the on-device model. Therefore, showing that it works fast and does not waste a lot of energy.

## 4. Experimental Evaluation

In this section, we evaluate the proposed method using a standard activity recognition chain [2] by comparing its performance with the traditional method, as shown in Table 1. We describe the designed and conducted the experiments, described the dataset collected, pre-process the data collected, build the recognition model, and evaluate it.

### 4.1. Experimental Setup

The participants were required to carry a waist-mounted Android smartphone, install the FahLog app on the phones, to select and record their activities from the list of predefined labels (depicted in Figure 6), get notifications, and submit data to our server. Each participant performs the experiments for 6 days. Table 1 shows the detail of the proposed method and traditional. We propose that if we give estimated activities using on-device deep learning inference as feedback to users through smartphone notifications, they can improve activity data collection. Therefore, to compare our proposed method with the traditional method, we created notifications on smartphones that displayed two different versions. Each version only differed in the user interface where the proposed method showed estimated activities using on-device deep learning inference when the device detects changes in the user’s activity. On the other hand, the traditional method showed messages “What are you doing?”, without estimated activities once every 15 min. We also request the users click the push notifications sent to assure that the users have seen the notifications. Each participant received both conditions, each of which showed three days. We randomly displayed the conditions for each participant to ensure that they were not affected by the day of experiments for each term. The participants were instructed with detailed instructions on how to do all process step by step using the same protocol provided. During data collection, the dataset was collected in the “wild” because the subjects provided data from their daily lives.

### 4.2. Data Description

The dataset was collected between June 2019, from six subjects within an age bracket of 25–30 years, performing one of six regular activities (as shown in the left column of Table 3) while carrying a waist-mounted Android smartphone that recorded the movement data (accelerometers in smartphones). Note that we requested them to carry a smartphone in the same position as the WISDM dataset used to train the on-device recognition model. As a result, we gathered 713 activity labels from all participants.

### 4.3. Activity Recognition Using Smartphone Sensors

Since we propose a standard activity recognition chain and a supervised learning approach for evaluations, we first preprocess the dataset collected and then evaluate it.

#### 4.3.1. Data Preprocessing

We put together the dataset by including three-axis accelerometer sensor data and the activity labels on the smartphones without clock and time synchronization because the sensor and the labeling system are both in the same device.

We used sliding windows of one minute with no overlapping. For each axis, average, standard deviation, maximum value and minimum value were extracted as features. An example of feature extraction is shown in Table 4. Before data proceeding, we excluded missing values. As a result, we obtained multivariate data of 9,129 samples with 12 variables for feature vectors.

Figure 8 shows the activity labels distribution of the data samples in our dataset. It is worth noting that the distribution was highly skewed, where some classes appeared more frequently than others. Since the imbalanced dataset can negatively influence the generalization and reliability of supervised learning algorithms, we employed the SMOTE algorithm: synthetic minority over-sampling technique as presented in [41] (an oversampling technique that creates new synthetic data samples in the minority classes, varying the features values of the existing data points based on their *k* nearest neighbors in the feature space) in order to balance our dataset. By upsampling the size of training and testing datasets separately.

#### 4.3.2. Evaluation Method

In this section, we present the effectiveness of the proposed method when we give estimated activities using on-device deep learning through smartphone notifications. The experiment was designed to test the performance of our classifier for a user-dependent scenario. In this case, the classifiers were trained and tested for each individual with her/his own data, and average accuracy and was computed. We show that the performance of several machine algorithms and LSTM have improvements with our method. We also show that the proposed method has improvements in the amount of data collected. To evaluate the proposed method using a technique of supervised learning algorithm for multiclass classification. We trained each participant separately using one deep learning classifier and several standard machine learning classifiers, including LSTM in the same way of the on-device model trained, logistic regression (LR) [42], linear discriminant analysis (LDA) [43], k-nearest neighbors (KNN) [44], decision tree (CART) [45], naive Bayes (NB) [46], support-vector machine (SVM) [47], and random forest (RF) [48].

To test the model’s ability we used stratified k-fold cross-validation. The folds are made by preserving the percentage of samples for each class to ensure each fold is a good representative of the whole. To account for label imbalance, the model performance was presented using the weighted average of precision, recall, F1-score of each class for the multiclass task. (i.e., averaging the support-weighted mean per label) So the average was weighted by the support, which was the number of samples with a given label. The “weighted” precision or recall score is defined in Equation (Equation 2). The same weighting is applied to F1-score.
(2)$$\frac{1}{\sum_{l\in L}\left|{\hat{y}}_{l}\right|}\sum_{l\in L}\left|{\hat{y}}_{l}\right|\phi \left(y_{l},{\hat{y}}_{l}\right)$$
*L* is the set of labels$\hat{y}$ is the true label*y* is the predicted label${\hat{y}}_{l}$ is all the true labels that have the label *l*$|{\hat{y}}_{l}|$ is the number of true labels that have the label *l*$\phi (y_{l},{\hat{y}}_{l})$ computes the precision or recall for the true and predicted labels that have the label *l*. To compute precision, let $\phi \left(A,B\right)=\frac{\left|A\cap B\right|}{\left|A\right|}$. To compute recall, let $\phi \left(A,B\right)=\frac{\left|A\cap B\right|}{\left|B\right|}$.

Note that weighted metrics is the performance of infrequent classes are given less weight since $|{\hat{y}}_{l}|$ will be small for infrequent classes. Therefore, weighted metrics may hide the performance of infrequent classes, which may be undesirable.

## 5. Results

Following the evaluation approach discussed above, we report our results of the validation together with a discussion of such results. We show the proposed method had improvements in data quality (the classification performance) compared to the traditional method. The average classification performance of all models results are shown in Figure 9. The F1-score performance results for each model are shown in Figure 10. The precision performance results for each model are shown in Figure 11. The recall performance results for each model are shown in Figure 12. The average classification performance results of all models for each user are shown in Table 5.

We also show that the proposed method has improvements in data quantity (the number of data collected) compared to the traditional method. Figure 8 shows the number of collected activity labels for both methods.

### 5.1. Quality of Collected Activity Data

Figure 9 shows F1-score, precision, and recall performance results of all machine learning models were improved with our proposed method compared to the traditional method. The F1-score was improved from 0.6240 to 0.7620 **(+0.138)** The precision was improved from 0.6440 to 0.7802 **(+0.136)** The recall of improved from 0.6366 to 0.7677 **(+0.131)**.

Figure 10 shows F1-score performance results of all machine learning models were improved with our proposed method compared to the traditional method. The F1-score of CART was improved from 0.657 to 0.770 **(+0.113)** The F1-score of KNN was improved from 0.667 to 0.801 **(+0.134)**. The F1-score of LDA was improved from 0.604 to 0.766 **(+0.162)**. The F1-score of LR was improved from 0.623 to 0.778 **(+0.155)**. The F1-score of LSTM was improved from 0.657 to 0.783 **(+0.126)**. The F1-score of NB was improved from 0.472 to 0.606 **(+0.134)**. The F1-score of RF was improved from 0.694 to 0.815 **(+0.121)**. The F1-score of SVM was improved from 0.623 to 0.775 **(+0.152)**.

Figure 11 shows precision performance results of all machine learning models were improved with our proposed method compared to the traditional method. The precision of CART was improved from 0.679 to 0.805 **(+0.126)**. The precision of KNN was improved from 0.665 to 0.793 **(+0.128)**. The precision of LDA was improved from 0.611 to 0.762 **(+0.151)**. The precision of LR was improved from 0.616 to 0.759 **(+0.143)**. The precision of LSTM was improved from 0.675 to 0.803 **(+0.128)**. The precision of NB was improved from 0.593 to 0.757 **(+0.164)**. The precision of RF was improved from 0.698 to 0.813 **(+0.114)**. The precision of SVM was improved from 0.619 to 0.738 **(+0.119)**.

Figure 12 shows recall performance results of all machine learning models were improved with our proposed method compared to the traditional method. The recall of CART was improved from 0.648 to 0.746 **(+0.098)**. The recall of KNN was improved from 0.681 to 0.814 **(+0.133)**. The recall of LDA was improved from 0.626 to 0.780 **(+0.154)**. The recall of LR was improved from 0.657 to 0.806 **(+0.149)**. The recall of LSTM was improved from 0.657 to 0.779 **(+0.121)**. The recall of NB was improved from 0.459 to 0.556 **(+0.097)**. The recall of RF was improved from 0.696 to 0.821 **(+0.137)**. The recall of SVM was improved from 0.677 to 0.833 **(+0.156)**.

Table 5 shows all users improve average F1-score, average precision, and average recall performances of all machine learning models with our proposed method compared to the traditional method.

### 5.2. Quantity of Collected Activity Data

Figure 8 shows the number of collected activity labels was increased with our proposed method. The number of activity labels increased from 311 to 402 **(+91)** compared to the traditional method.

Table 6 shows the number of labels of each activity class by comparing the proposed and traditional. While some activity classes have more labels with the proposed method, only one class has fewer labels with the proposed method.

The number of walking labels was increased from 112 to 135 **(+23)**. The number of upstairs labels was increased from 13 to 14 **(+1)**. The number of standing labels was increased from 68 to 85 **(+17)**. The number of sitting labels was increased from 101 to 148 **(+47)**. The number of downstairs labels was increased from 16 to 20 **(+4)**. The number of jogging labels was decreased from 1 to 0 **(−1)**.

## 6. Discussion and Future Directions

By evaluating the dataset and comparing with the traditional method, the results reflect that our proposed method has improvements in data quality (the performance of a classification model) for all machine learning models evaluated and data quantity (the number of labels collected) that indicate improvements in activity data collection. What we have found most interesting is that all users improve quality of activity data collection with the proposed method, as shown in Table 5. While RF achieves the highest F1-score at 81.5%, LDA has the most improvements by 16.2%. RF achieves highest the precision at 81.3%, NB has the most improvements by 16.4%. SVM achieves highest the recall at 83.3% and also has the most improvements by 15.6%. While this study enabled us to improve activity data collection effectively, there are some limitations that we would like to point out and reference in the future.

First, while we notified information about estimated activity when the user is currently doing the activity, it might be necessary to design both our mobile app and our recognition model to identify when a user starts or stops a particular activity, such as walking, biking, or driving (e.g., detect when users start and end an activity). For activity recognition systems, it is crucial to collect correct segments data. In other words, we need a labeled sequence of activities (i.e., the start and finish times of the events). Hence, if the app can be used to detect changes in the user’s activity, we can also deliver this information as feedback to the user for better activity data collection. Researchers may consider this idea for other purposes, for example, an app subscribes to a transition in activities of interest and notifies the user only when needed (e.g., the app notifies driving when a user starts driving and mute all conversations until the user stops driving).

Second, we used the WISDM dataset to train our deep learning model. Hence, the smartphone’s position is limited for activity data collection in our experiment as we have to put the smartphone in the same position. If the smartphone’s position and/or orientation is discrepant from theirs, the on-device inference will not be correct. Consequently, considering to collect training dataset by ourselves will be vital. Also, we can collect more data to make the samples balance among activity classes. Furthermore, while we applied a three-axis accelerometer for training the recognition model and inferring on a smartphone device, other smartphone sensors would be useful for more accurate recognition. For example, adding gyroscope can help indicate orientation. We will leave this for future work.

Third, we run the trained model on a device without retraining. When designing activity recognition (machine learning) systems, it is crucial to understand how our data is going to change over time. A well-architected system should take this into account, and a plan should be put in place for keeping our models updated. There are several ways to retain the model, for example, manual retraining by training and deploying your models with fresh data using the same process you used to build your models in the first place or continuous learning by using an automated system to evaluate and retrain your models continuously (e.g., hosting a model on the cloud [49]). However, retraining the model to maintain machine learning systems would be challenging for research questions in future work, for example, how do we ensure our predictions continue to be accurate? Similarly, how do we keep our models up-to-date with new training data?

Fourth, as our proposed method can be applied for several algorithms, but the main on-device inference model that drove our work—that LSTM-based deep learning model. If a training model were evaluating using other deep learning methods, such as CNN, CNN + LSTM, then there would be value in expanding—why LSTM? What are the challenges that are different from other methods? Which method is best?

Finally, we plan to evaluate the method with long-term data collection and more diverse samples, find data insights as well as find out the correlations between accuracy, the number of activity labels and classes to show whether and how strongly pairs of variables are related. For example, do notifications affect the number of activity labels or do notifications affect the number of activity classes? Answering these questions, it would also be helpful to understand user motivations and support activity data collection further. Likewise, we have seen that although the number of activity labels is increased with our method, not all activity classes (see in Table 6). Therefore, analyzing the data collected more deeply will be useful to understand correlation and causation.

## 7. Conclusions

We have proposed a method to use on-device deep learning inference to detect activities that users are doing as feedback for optimizing activity data collection in smartphone-based activity recognition. The proposed method was validated with mobile sensors and 713 activity labels that we collected from 6 participants. By evaluating with the dataset, the preliminary results indicate that our proposed method has improvements in F1-score, precision, and recall for all machine learning classifiers compared to the traditional method. Moreover, the amount of data collected has increased with the proposed method. There are several challenging areas that we see as ripe for next steps, for instance, exploiting on-device deep learning inference for detecting changes in the user’s activity, collecting own training data for on-device inference model, adding more sensor types for training activity recognition models, retraining an on-device model, showing and comparing with other deep learning methods as well as collecting more data and analyzing it deeply. We hope that this work can spark future studies of on-smartphone deep learning as well as other edge devices that will be useful for data collection in activity recognition systems.

## Figures and Tables

**Figure 1 sensors-19-03434-f001:**
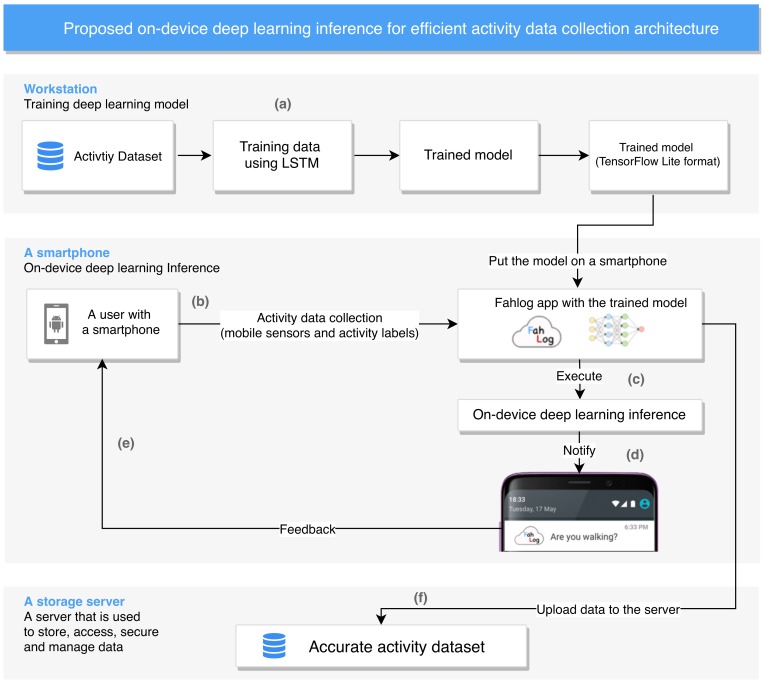
The system architecture of the proposed on-device deep learning inference for efficient activity data collection works as following (**a**) we first train activity labels with mobile sensors using an long short-term memory (LSTM) for recognition model and deploy it for on-device inference, (**b**) we collect mobile sensors and activity labels on a smartphone from a user, (**c**) the smartphone detects an activity that the user is doing by using an on-device deep learning inference model adopted, (**d**) the user receives information about the estimated activity as feedback (e.g., the notification is showing that “are you walking?” means the device is on a user who is walking), (**e**) the user repeats the process of activity data collection efficiently, and (**f**) we finally obtain accurate activity dataset.

**Figure 2 sensors-19-03434-f002:**
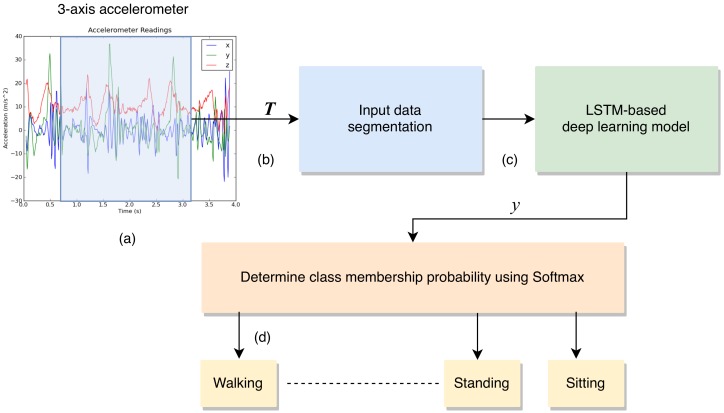
A schematic diagram of the proposed LSTM-based deep learning model for activity recognition system works as following (**a**) the inputs are raw signals obtained from acceleration sensors, (**b**) segment into windows of length *T*, (**c**) fed into LSTM-based deep learning model, (**d**) and finally, the model outputs class prediction for each time step.

**Figure 3 sensors-19-03434-f003:**
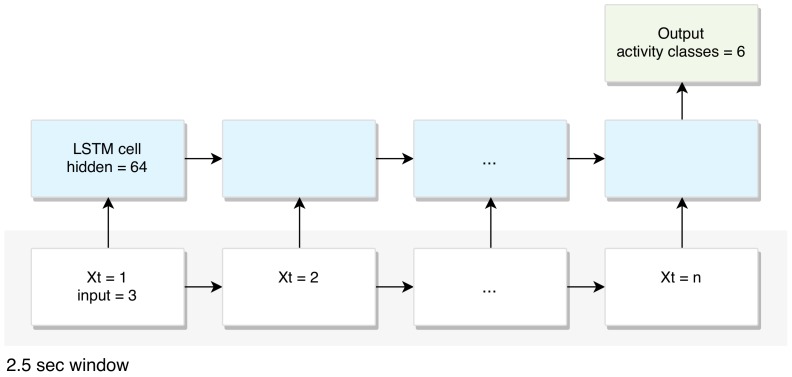
Many-to-one long short-term memory (LSTM) network architecture used for activity classification with six classes. n stands for the number of samples included in a 2.56 s window.

**Figure 4 sensors-19-03434-f004:**
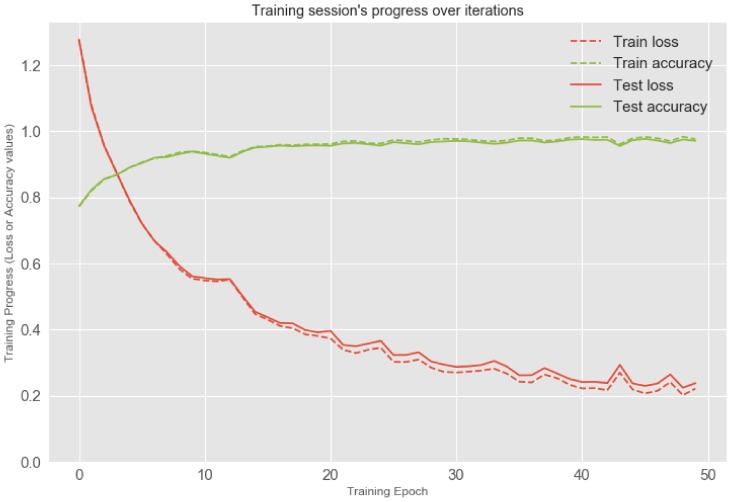
Training session’s progress over iterations.

**Figure 5 sensors-19-03434-f005:**
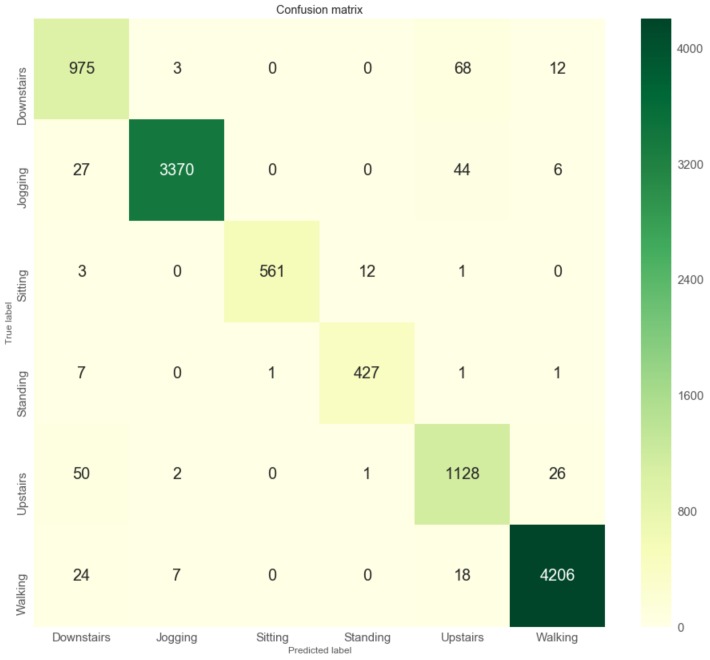
The results for a classifier of the LSTM model.

**Figure 6 sensors-19-03434-f006:**
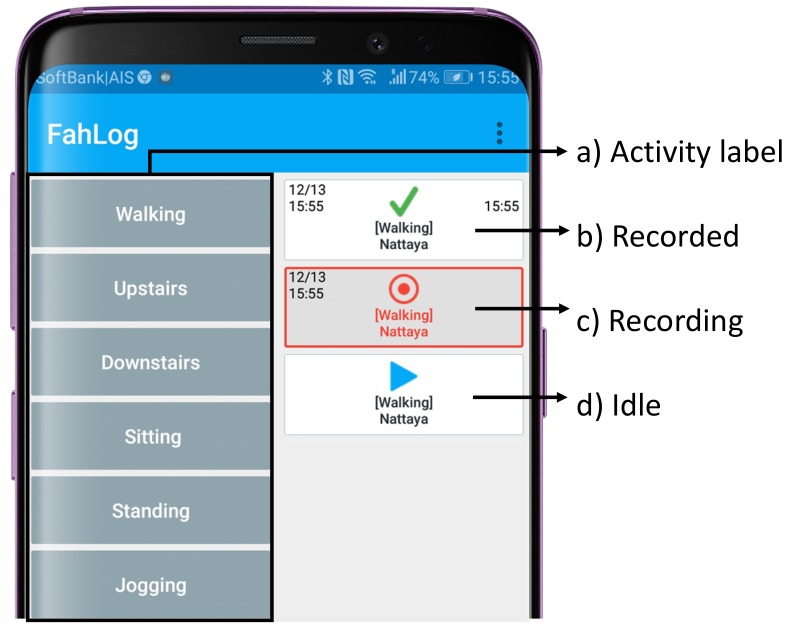
FahLog: a mobile app for collecting sensor data and activity labels.

**Figure 7 sensors-19-03434-f007:**
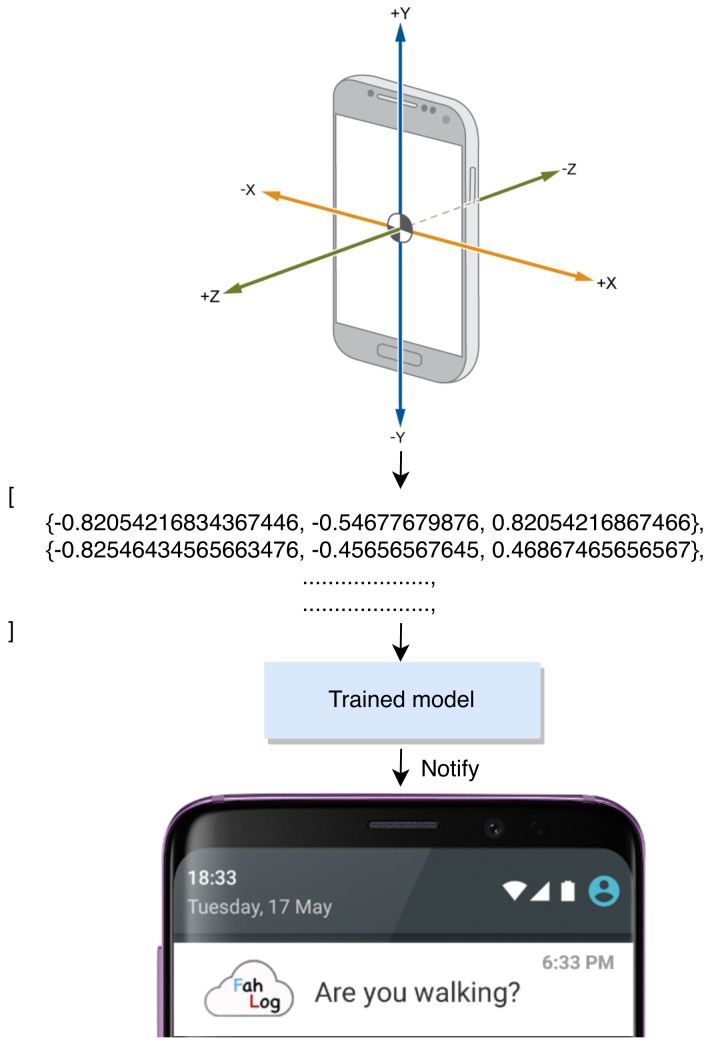
Steps to show a estimated activity as feedback to a user.

**Figure 8 sensors-19-03434-f008:**
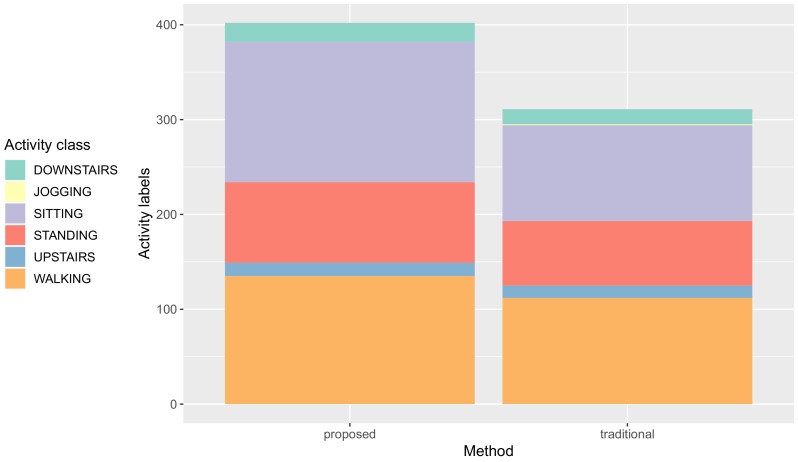
The number of activity labels for each method.

**Figure 9 sensors-19-03434-f009:**
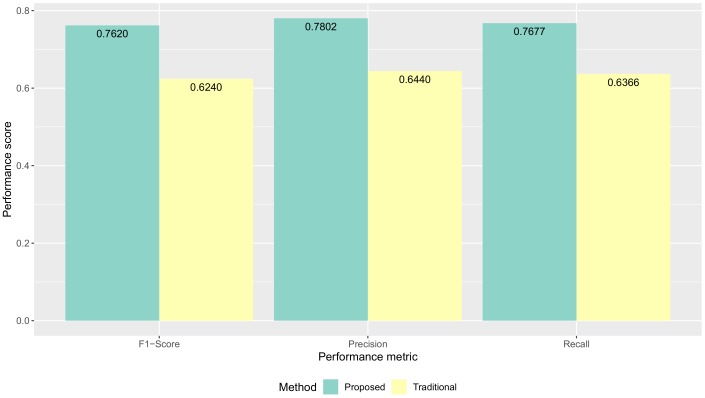
The average classification performance of all models for each method.

**Figure 10 sensors-19-03434-f010:**
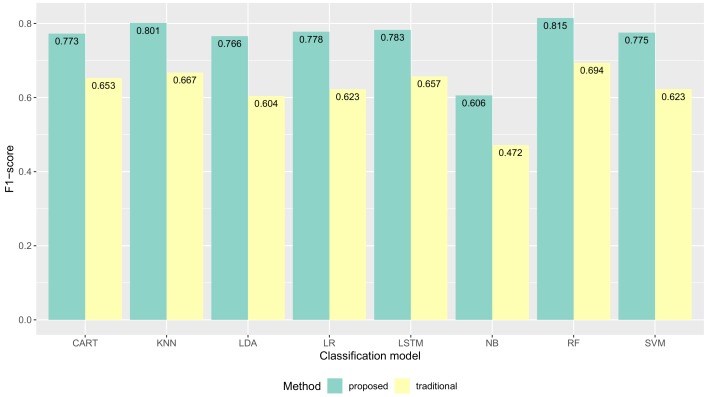
The F1-score performance results of several machine learning models.

**Figure 11 sensors-19-03434-f011:**
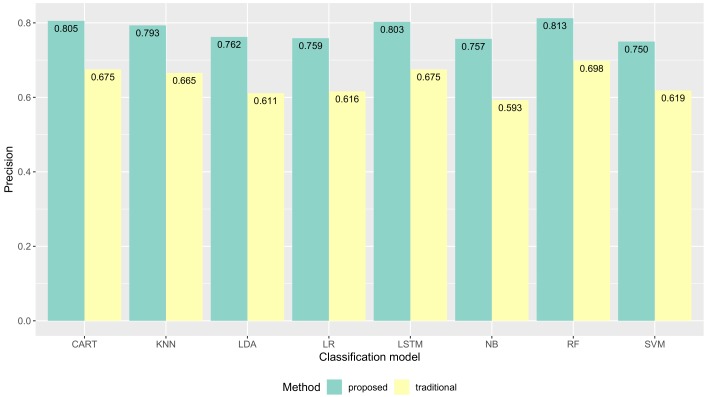
The precision performance results of several machine learning models.

**Figure 12 sensors-19-03434-f012:**
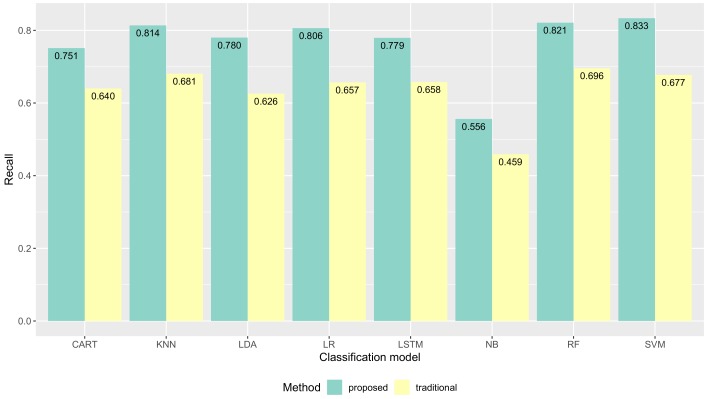
The recall performance results of several machine learning models.

**Table 1 sensors-19-03434-t001:** Experimental design.

Method	Conditional Detail
Proposed	Receive notifications of estimated activities using on-device deep learning inference
Traditional	Receive notifications with messages “What are you doing?” without estimated activities.

**Table 2 sensors-19-03434-t002:** Some core parameter definitions for the training.

Parameter	Value
LSTM layer	2 fully-connected
epochs	50
hidden layer units	64
output classes	6
input features per timestep	3 ($acc_{x},acc_{y},acc_{z}$)
timesteps per series	200
learning rate	0.0025
batch size	1024

**Table 3 sensors-19-03434-t003:** The number of activity labels collected.

Activity Class	# labels
Walking	247
Jogging	1
Sitting	249
Standing	153
Downstairs	36
Upstairs	27
**Total**	**713**

**Table 4 sensors-19-03434-t004:** An example of feature extraction.

Feature	Value
$mean_{x}$	num −0.82054216867469876 …
$max_{x}$	num −0.622 …
$min_{x}$	num −1.207 …
$sd_{x}$	num 0.085123482931909022 …
$mean_{y}$	num 1.3659708029197057 …
$max_{y}$	num 5.468 …
$min_{y}$	num −4.118 …
$sd_{y}$	num 1.1740472194146572 …
$mean_{z}$	num 9.819719626168224 …
$max_{z}$	num 9.909 …
$min_{z}$	num 9.742 …
$sd_{z}$	num 0.894526836753883 …

**Table 5 sensors-19-03434-t005:** The average classification performance of all models for each user.

User	Method	F1-Score	Recall	Precision
98	proposed	0.7778	0.7756	0.7973
98	traditional	0.7127	0.7139	0.7391
98	**Improvement**	**+0.0651**	**+0.0616**	**+0.0582**
99	proposed	0.7009	0.7119	0.7156
99	traditional	0.4442	0.4830	0.4605
99	**Improvement**	**+0.2567**	**+0.2289**	**+0.2551**
101	proposed	0.8700	0.8774	0.8727
101	traditional	0.6449	0.6619	0.6701
101	**Improvement**	**+0.225**	**+0.215**	**+0.203**
103	proposed	0.7693	0.7663	0.7950
103	traditional	0.6490	0.6685	0.6584
103	**Improvement**	**+0.120**	**+0.098**	**+0.137**
104	proposed	0.7881	0.7954	0.8120
104	traditional	0.6333	0.6223	0.6705
104	**Improvement**	**+0.155**	**+0.173**	**+0.142**
105	proposed	0.6658	0.6794	0.6888
105	traditional	0.6600	0.6702	0.6654
105	**Improvement**	**+0.006**	**+0.01**	**+0.023**

**Table 6 sensors-19-03434-t006:** The number of activity labels of each activity class for each method.

Activity Class	Proposed	Traditional	Improvement
Walking	135	112	**+23**
Upstairs	14	13	**+1**
Standing	85	68	**+17**
Sitting	148	101	**+47**
Downstairs	20	16	**+4**
Jogging	0	1	**−1**
**Total**	**402**	**311**	**+91**

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
