# Peer review of "On-Device Deep Learning Inference for Efficient Activity Data Collection"

_sensors, 2019, doi:10.3390/s19153434_

Round 1

Reviewer 1 Report

This paper focuses on performing on-device annotation of activity data.  

Authors choose an interesting research question and on-device cloud undefended analysis is an interesting trend in the community.

However, it suffers from major shortcomings which limit its contribution. Following lists major short comings that has been identified. Authors should address these issues in the next version of their paper.

  The first paragraph of the introduction can be written in three sentences and it is too much waste of time and space. It is not necessary to use many sentences + references to mention that manually annotation of the data is expensive.

Introduction lacks a solid  discussion about the need for  (i) on-device machine learning and DOI: 10.1109/TKDE.2016.2592527, DOI: 10.1109/TMM.2013.2265674, doi>10.1145/1149488.1149490, (ii) deep learning revolution in the field of machine learning and its impact 10.1109/MNET.2018.1700202, https://doi.org/10.1016/j.neunet.2014.09.003

, DOI: 10.1109/MCE.2016.2640698. Instead it focuses has some text about smartphone activity recognition, which is not novel and existed in the community more than 10 years. Then it jumps to tensor flow.

I don’t believe another dataset of activity recognition might be helpful. There are many datasets available and as it has been stated previously there is not much to deliver to the community as a dataset contribution. Nevertheless, this attitude of authors should get some appreciation.

Authors claimed in their title that employ on-device machine learning. That is an interesting research direction. However, they failed to acknowledge several visionary works in this area before the release of TF lite, including NoCloud DOI: 10.1109/MPRV.2018.011591063, edge Computing DOI: 10.1109/JIOT.2016.2579198.… In particular, most of the related work discussion is limited to TF tool. The reader didn’t get anything useful from this section.

Manual annotation of personal data is cumbersome. Authors provide a manual annotation tool in their framework. They should describe whether such an annotation will exited in the real-world product or it is only for this research work. If it is limited to the research work, how can they extend this into a real-world approach?

to build the model authors employ an existing dataset, but in the introduction they have explained that they introduce their own dataset. Authors should clarify this inconsistency in their paper. In particular, why another dataset for model building and your own dataset for testing the model?

There is no need for the author to write the formula of softmax. Any reader of the paper who has small familiarity with deep learning should know this function.

There is not information about demography of participants and also no information about the number of participants or duration of the experiment reported in section 3.2. Please resolve this too.

Please consider to revise the title of section 3.3. it is too long and the reader didn’t understand the need for prediction here. To my understand this work is focused on recognition and not prediction. 

Authors should provide a comparison between their approach and state-of-the-art method or better existing API in the market such as Google Play, not just traditional machine learning algorithms. Moreover, it is obvious that deep learning outperforms traditional models and such an evaluation doesn’t make much sense.

Also there is no report about the response time and energy use of the method proposed by authors in comparison to state-of-the-art methods. Please provide that evaluation as well.

Author Response

I uploaded a PDF file for the revised version and a point-by-point response to the reviewer’s comments.

Reviewer 2 Report

This paper presents a solution to improve the collection and annotation of sensor data for the purpose of Activity Recognition. The paper applies a deep learning approach through LSTM which is highlighted as the novel contribution. The paper tackles and interesting and increasingly important problem of collecting large datasets in the wild.

Whilst the paper is generally well written it does suffer from a number of limitations in how the methodology is presented and the review of the literature. In particular, the methods section does not communicate clearly how the proposed labeling approach is being tested and what it is being compared too. Furthermore there seems to be a disconnect between the LSTM approach and the evaluation. Results focus mainly on Supervised learning approaches and does not consider LSTM which was highlighted as the novel element of this work. Additionally, a review of other approaches to data labeling has not been provided. This makes it difficult to asses the true novelty of this solution. Details of some of these papers are provided below. Change point detection has been used to better identify transitions in activities.

The paper requires substantial changes to the methods and how they are described. There are also some changes to the methodology, in terms of validation that could be made. Specific changes are provided below.

Abstract- It would be good to have more detail here on what evaluation was undertaken, what was the comparison and what quantitative results are achieved.

Line 32- This sentence doesn’t make sense and needs re-written

Line 34- needs rewritten for English language.

Page 2- Para starting line 36. This sets the scene of the paper, however doesn’t clearly outline what is achieved. For example, do you mean that the user will be prompted to provide labels for uncertain activities? It isn’t clear if this has been deployed or is used retrospectively on previously collected data.

Line 76- can you add a reference here to validate the benefits of LSTM

There is a lot of work being carried out in the area of data annotation, I think a review of this is missing, although it is alluded to in the introduction. The percom workshop arduous will give some ideas here as well as https://pdfs.semanticscholar.org/78ff/53889a6f6abde3ffa43b9e69050bfa31f1f6.pdf https://www.eecs.wsu.edu/~cook/pubs/tkde18.pdf

Figure 1. is there some retraining here to personalize the model? Is the final layers of the model retrained? It would be nice to see how the loop is closed here.

Line 125, when mentioning the open dataset here I would provide a link to the WISDM

Line 131, what are the 6 attributes?

143 what is the F measure here?

How does the imbalance in the data set impact upon the results.

How was the data split into train and test?

Figure 5 is difficult to read and doesn’t really provide any insight. Perhaps it should be removed. It is also not mentioned in the text.

165, “then we take the compressed “version” of …

175: title is missing words? “To provide “e.g. Information, Insight”

For the experiment 12 activities are targeted, however, only 6 are trained using the open dataset? How is this accounted for, or what impact does this have on the accuracy of the model?

The F1 is used to validate the classifiers. I think this should also be applied to the LSTM.

Table 2 isnt well explained. What is the traditional method? is it experience sampling or offline? Is it Fahlog vs the prompted labelling? What about experience sampling?

Was data collected in the wild or through a protocol. How do you know reported labels are correct?

Features extracted are very simple and don’t seem to cover the frequency domain. What was the rational for this?

Why was a 1 second window presented?

LSTM should be used in the evaluation of results.

How does the supervised machine learning method link to the LSTM processing, why are these compared? Should the comparison not focus on the annotation procedures?

Smote used to rebalance, how does this impact on the results?

The validation approach has been shown to artificially inflate results. Consider rerunning analysis using a leave one out validation method or a hold out. https://arxiv.org/pdf/1904.02666.pdf

Line 279 How does the solution improve data quality?

Author Response

(The authors gave the same response as above.)

Round 2

Reviewer 1 Report

I believe authors has incorporated most of my recommendations, not all of them. However, it is ok, they spent significant amount of time to improve their paper and respect reviewers' concerns.

Therefore, I would vote for accept

Reviewer 2 Report

I would like to thank the authors for addressing my concerns in the current version of the report. There are a number of small typos/ English mistakes. Hopefully these will be picked up in editing.